# STUPD: A Synthetic Dataset for Spatial and Temporal Relation Reasoning

## Abstract

Identifying relations between objects is crucial for understanding the semantics of a visual scene. It is also an essential step in order to bridge visual and language models. However, current state-of-the-art computer vision models still lack the ability to perform spatial reasoning well. Existing datasets mostly cover a relatively small number of spatial relations, all of which are static relations that do not intrinsically involve motion. In this paper, we propose the **S**patial and **T**emporal **U**nderstanding of **P**repositions **D**ataset (STUPD) – a large-scale video dataset for understanding spatial and temporal relationships derived from prepositions of the English language. The dataset contains 150K visual depictions (videos and images), consisting of 30 static and dynamic spatial prepositions, in the form of object interaction simulations generated synthetically using Unity3D. In addition to spatial relations, we also propose 50K visual depictions across 10 temporal relations, consisting of videos depicting event/time-point interactions. To our knowledge, no dataset exists that represents temporal relations through visual settings. In this dataset, we also provide 3D information about object interactions such as frame-wise coordinates, and descriptions of the objects used. The goal of this synthetic dataset is to help models perform better in visual relationship detection in real-world settings. We demonstrate an increase in the performance of various models over 2 real-world datasets (ImageNet-VidVRD and Spatial Senses) when pretrained on the STUPD dataset, in comparison to other pretraining datasets.

## 1 Introduction

Identifying relationships between objects is crucial for semantic understanding of the visual world. However, current state-of-the-art computer vision models still find it challenging to understand relationships (Conwell & Ullman, 2022; Cho et al., 2022; Liu et al., 2021; Thrush et al., 2022; Ramesh et al., 2022; Zhang et al., 2022). For instance, even for simple relations in 2D pixel space such as "left", "right", "above" and "below", Cho et al. (Cho et al., 2022) found a large gap between upper-bound accuracy and the performance of generative transformers. Compared to an upper-bound accuracy of 99.3%, the average accuracy of 3 models was only 24.7%, with the best model achieving 51.2%.

In human languages, relational concepts are conveyed using prepositions, which are words used *"to show a relationship in space or time"* (Litkowski & Hargraves, 2021a). Examples of prepositions include "above", "before" and "with". Existing computer vision datasets cover English parts-of-speech such as nouns/objects (Deng et al., 2009a; Krizhevsky et al., 2009), verbs/actions (Sigurdsson et al., 2016; Ji et al., 2020; Kay et al., 2017), adjectives/attributes (Parikh & Grauman, 2011; Krishna et al., 2016), etc. However, despite their importance, prepositions are significantly understudied in computer vision as a distinct class of concepts.

Prepositions may have one or more senses, which are distinct definitions of a word in different contexts. For example, the preposition "against" has 2 distinct spatial senses (Litkowski & Hargraves, 2021a). One refers to a situation where 2 objects are moving in opposite directions and the other where an object is leaning on another. For simplicity, we will henceforth use the term "preposition" to refer to both prepositions (the words) and their senses (the definitions), except where clear distinctions are required. A detailed glossary of all terms introduced in this paper is included in the Appendix.

From Table 1, it can be observed that image datasets that contain hundreds to thousands of relation classes actually have fewer than 30 prepositions (an exception is the recent VSR dataset (Liu et al.,

2022) which covers 65 prepositions). As for existing video datasets, only 6-8 prepositions are covered. Furthermore, datasets thus far contain only *static* prepositions, which are prepositions that do not necessarily involve any motion, such as "above" and "behind". The vast majority of such examples come from very simple and intuitive preposition classes such as "on" or "near", which are easier to label by human annotators. None of the existing datasets include *dynamic* prepositions, which are prepositions that intrinsically involve motion, such as "into", "onto", etc. Finally, existing datasets are also extremely imbalanced due to the long-tailed distribution of relationship occurrences.

This kind of highly restrictive relational domain in existing datasets is not an effective approach towards visual reasoning, because it only focuses on position, while ignoring many fundamental relational characteristics, such as relative speed, contact and physical forces of interaction. The prospect of the ability to distinguish between different spatial (as well as temporal) configurations with higher granularity, thus, makes it worthwhile to study the wider variety of prepositions for effective visual reasoning. Through this, datasets can be richer in information, and models would be able to differentiate between many related but different relational categories (such as "above" and "over"). A granular understanding of prepositional relations also allows for better understanding of language semantics, which is an equally important and complementary aspect of visual reasoning in the understanding of a scene.

Apart from spatial reasoning, understanding temporal relations is also a crucial component for visual reasoning. Many relations require understanding dynamics of interactions over time. Visual representation of temporal relationships is a challenging task because temporal concepts are unintuitive to visualize. This is one of the reasons why temporal relations are heavily underrepresented in visual reasoning datasets. Without effectively understanding temporal relations, spatial relations remain isolated, and their progression cannot be understood. Thus spatial and temporal relations should be treated as equally important aspects of visual reasoning.

**Contributions.** To address these issues, we created the Spatial and Temporal Understanding of Prepositions Dataset (STUPD) – the first dataset to include dynamic spatial prepositions and temporal relations. The contributions of this paper are as follows:

1. **Comprehensive synthetic dataset for spatial relations**: This paper introduces a dataset consisting of 150,000 images and videos that capture 30 different spatial relations. The dataset incorporates physical interactions using a sophisticated physics engine coupled with diverse backgrounds.

2. **Comprehensive synthetic dataset for temporal relations**: In addition to the spatial relations dataset, this paper introduces a separate dataset comprising 50,000 sets of videos depicting 10 different temporal relations. Through this, the paper also introduces a definitive framework for defining and distinguishing between different temporal relations, for future works to build on.

3. **Detailed 3D information and bounding box annotations**: To enhance the quality and usability of the dataset, each image and video in the dataset is accompanied by detailed 3D information and bounding box annotations.

4. **Effective pre-training dataset with real-world applicability**: The proposed datasets are primarily designed to serve as a highly effective pre-training resource for computer vision models. Pre-training on this dataset provides a solid foundation for subsequent fine-tuning on real-world datasets. Later in the paper, we demonstrate that pretraining on STUPD increases performance on real-world visual reasoning tasks.

## 2 RELATED WORK

### 2.1 IMAGE DATASETS

In recent years, image-based datasets have attempted to present spatial relationships through simple 2D object interactions (Yang et al., 2019; Krishna et al., 2016; Lu et al., 2016). However, 2D interactions restrict the scope of distinguishable visual relations. Synthetically generated datasets are becoming increasing popular as a way to bridge the information gap in image datasets through 3D spatial relations (Johnson et al., 2017; Liu et al., 2021). An example is the CLEVR dataset (Johnson et al., 2017) which consists of synthetic images with objects arranged in various configurations to promote generalization and systematic reasoning.

| Type | Dataset | Year | 3D info? | # Preps | Dyn? | Tem? | Real/ Synth | Size |
|------|---------|------|----------|---------|------|------|-------------|------|
| Image | VSR (Liu et al., 2022) | 2022 | N | **65** | N | N | Real | 10K |
| Image | Liu et al. (Liu et al., 2021) | 2021 | N | 6 | N | N | Synth | 83K |
| Image | Rel3D (Goyal et al., 2020) | 2020 | Y | 25 | N | N | Synth | 27.3K |
| Image | SpatialSense (Yang et al., 2019) | 2019 | N | 9 | N | N | Real | 11.5K |
| Image | CLEVR (Johnson et al., 2017) | 2017 | N | 4 | N | N | Synth | 100K |
| Image | Visual Genome 50 (Xu et al., 2017) | 2017 | N | 21 | N | N | Real | 108K |
| Image | VRD (Lu et al., 2016) | 2016 | N | 24 | N | N | Real | 5K |
| Image | Scene Graphs (Johnson et al., 2018) | 2015 | N | 29 | N | N | Real | 5K |
| VR | iGibson 2.0 (Li et al., 2021) | 2021 | Y | 6 | N | N | Synth | N/A |
| Video | CATER (Girdhar & Ramanan, 2020) | 2020 | N | 7 | N | Y | Real | 5.5K |
| Video | Action Genome (Ji et al., 2020) | 2020 | N | 6 | N | N | Real | 1.75K |
| Video | VidOR (Shang et al., 2019) | 2019 | N | 8 | N | N | Real | 10K |
| Video | **STUPD (ours)** | 2023 | Y | 40 | Y | Y | Synth | **200K** |

Table 1: Comparison of relations datasets. (Preps = Prepositions (relations), Dyn = Dynamic in nature, Tem = Temporal, Synth = Synthetically generated)

However, synthetic datasets in this domain do not provide three-dimensional information about object location or orientation, rendering the perceptual input provided as effectively two-dimensional. Some works such as Goyal et al. (2020) provide annotated synthetic 3D scenes. This allows models to better understand object interactions and distinguish between subtle visual relations such as impact and contact.

A common theme across different visual relation datasets is to mix complex actions and prepositional relations (Goyal et al., 2017). For instance, in the Action Genome dataset (Ji et al., 2020), the action "sitting" and preposition "on" is combined into a single dynamic relation "sitting on a sofa". However, actions themselves require a fundamental understanding of spatial relations, as put forth by Hua et al. (2022), who argue that actions can be decomposed into chains of consecutive spatial relations between objects. Hence, relation understanding tasks should sit at the root of all other tasks that involve understanding more complex spatio-temporal relationships. Similarly, many datasets (Johnson et al., 2018; Liu et al., 2022) present a larger number of spatial relations, which are overlapping in meaning. For example, "below" and "beneath", or "adjacent to" and "beside". Both pairs includes different prepositions but are essentially describing the same preposition sense. Hence, the mixing of spatial relations with similar meanings results in redundant representations.

### 2.1.1 GRAPH-BASED SCENE RELATION REPRESENTATION

Relations can be explicitly modeled as graphs (Ashual & Wolf, 2019; Ji et al., 2020; Krishna et al., 2016; Xu et al., 2017; Johnson et al., 2018), which can substitute the need for 3D information in a restricted manner. This form of representation can also allow multiple spatial relations to co-exist, which may be useful in understanding complex scenes. While these works have shown strong performance in identifying low-level object relationships, understanding of higher-order relationships are still not clearly understood through this approach.

### 2.2 VIDEO DATASETS

Many spatial relations have a dynamic nature, meaning that they intrinsically involve motion (e.g. "onto"), which cannot be represented by image datasets. Various works have proposed video datasets (Li et al., 2021; Ji et al., 2020; Shang et al., 2019), but they only cover the basic static positional prepositions (e.g. "behind", "above" and "below"). Shang et al. (2019) have a few additional static prepositions such as "(facing) towards", but overall, dynamic spatial and also temporal prepositions are severely under-researched. The CATER dataset (Girdhar & Ramanan, 2020) covers just the 3 most basic temporal prepositions ("before", "during" and "after").

### 2.3 OTHER RELATED VISUAL REASONING TASKS

Various other tasks are related to visual relationship reasoning, which require the use of both spatial and temporal cues to match visual features with labels for objects and relations. This includes tasks such as video grounding (Su et al., 2021; Li et al., 2022; Zeng et al., 2020) and visual question answering (Antol et al., 2015; Yusuf et al., 2022). Hence, many methods from visual relationship reasoning can be transferred to the above mentioned tasks and vice versa.

# 3 THE STUPD DATASET

The STUPD dataset is a dataset for visual reasoning. It contains synthetically generated images and videos depicting spatial and temporal relations between objects. These relations are derived from the list of prepositions of the English language, which are words representing relations between different subjects within a sentence. The STUPD dataset provides 5,000 images/videos for each preposition, resulting in 150,000 images and videos corresponding to spatial relations (referred to as **Spatial-STUPD**) and 50,000 collections of videos corresponding to temporal relations (referred to as **Temporal-STUPD**). The videos contains realistic interactions between objects of different kinds. The dataset is statistically balanced with respect to object combinations. The dataset can be used to pretrain models to perform visual reasoning better, and we demonstrate this in the paper.

## 3.1 PREDICATE VOCABULARY

The Prepositions Project (TPP) (Litkowski & Hargraves, 2021b), a database of all prepositions in the English language, lists 373 prepositions in total. We use TPP as the source for our vocabulary, and select prepositions only from the two largest groups in TPP (spatial and temporal prepositions) for this paper. We first apply a structured filtering process on the list of all prepositions from TPP, the details of which are outlined in the appendix. Through the filtering process, we shortlisted **30 spatial prepositions and 10 temporal prepositions**. These prepositions act as predicate relations for our visual reasoning dataset. Spatial relation categories are divided into two subcategories – static spatial relations (relations that do not involve relative movement between objects) and dynamic spatial relations (relations that involve movement of the objects). We describe all these relation categories, along with their definitions and context of usage, in Appendix 7.

## 3.2 SETTING AND STRUCTURE

### 3.2.1 SPATIAL DATASET STRUCTURE

Consider a spatial relation triplet <*subject, predicate, object*>. For each predicate (relation) in the STUPD dataset, *subject* and *object* are represented by a collection of 3D objects. These 3D object templates (also referred to as prefabs) were selected from the ShapeNet dataset (Chang et al., 2015), which contains high-quality annotated 3D templates of real-life objects. The detailed curation process of the prefabs used is explained in Appendix A.3.

We group all object categories into 8 supercategories based on size and physical properties to simplify the types of interactions between different objects. These 8 supercategories are *small objects* (everyday objects that are small enough to be easily maneuvered), *furniture, vehicles, person, large-scale grounded objects* (large heavy objects that are usually grounded, e.g. buildings)*, containers, track* (roads and paths), and *tunnels*. The idea behind supercategories is that categories within a supercategory have similar behavior of physical interaction.

Overall, we curated 183 prefab instances varying across 45 object categories and 8 supercategories. An overview of the 3D prefabs used, along with other design choices are presented in Appendix A.3.

It should be noted that in the STUPD dataset, the representation of relation triplets *(subject, predicate, object)* has a slightly different meaning than in previous works. Certain predicate relations in our vocabulary such as *(moving) up* and *(scattered) all over (the scene)* describe the relation between the predicate and the subject category (and not any object category). Hence, the *(object)* is empty for certain spatial relation categories. Note that subjects as well as objects can refer to multiple instances of the same category of prefabs.

### 3.2.2 TEMPORAL DATASET STRUCTURE.

Temporal predicates in the STUPD dataset depict a relation between 2 events (a stretch of time where something occurs) or time points (a single moment of time). Consider the temporal relation triplet <*Event/TimePoint A, relation, Event/TimePoint B*>. The challenging part of visual temporal relation representation is the visual depiction of events and time points. In this dataset, temporal relations are represented by means of videos, where events and time points are depicted using the spatial dataset generated. Each event is represented by a spatial relation (static or dynamic) that occurs over variable time spans. A static event simply means that there is an occurrence of a static relation a certain number of frames. On the other hand, time points are represented by single frame

inside the temporal videos, and these are sampled from only static spatial events, since a single frame cannot represent the temporal nature of a dynamic spatial relation.

### 3.3 DATASET CHARACTERISTICS

#### 3.3.1 SPATIAL-STUPD DATASET CHARACTERISTICS

All static spatial relations are generated as single RGB images(frames)($f = 1$), while dynamic spatial relations are generated as a collection of $f = 30$ consecutive RGB images (frames), which can be combined together to create a video depicting object interactions with dynamic movement. We synthetically generate 5,000 examples of each spatial relation using the Unity3D perception platform ((Borkman et al., 2021)), which allows the use of a physics engine to emulate realistic physical interactions between different objects.

To ensure enough variance in the dataset, we randomize a variety of parameters of the generated images, such as the selection of the objects (in a constrained manner to only allow selective super-category interactions, described above), the color of the objects, the distance between the objects, the relative position and rotation of the objects, the perspective of the camera, and even the background of the image. All visual relations in the STUPD dataset are with respect to the camera's perspective, hence removing any ambiguity of perspective. We provide annotations for each spatial interaction in the form of subject/object information (including category, supercategory, bounding box information and 3D coordinates), as well as the predicate relation category. Note that all spatial relations are independent of each other. Hence each spatial interaction corresponds to only one predicate relation category. Some examples of our dataset can be seen in Figure1 as well as in Appendix figures 6 and 7.

#### 3.3.2 TEMPORAL-STUPD DATASET CHARACTERISTICS

We generate pairs of videos of a constant length of $W = 150$ frames (referred to as the temporal window), where each video corresponds to the occurrence of a single event or time point. An important characteristic of temporal relations is the overlapping nature of temporal relation predicates. Event/TimePoint interactions can represent multiple temporal relations simultaneously. For example, consider *Event A* which occurs just after *TimePoint B*. In this case, temporal triplets *<Event A, after, TimePoint B >* and *<Event A, around, TimePoint B>* both apply. Hence in the STUPD dataset, each temporal interaction may have multiple temporal relation categories associated. An overview of all temporal relations is presented in Figure 2.

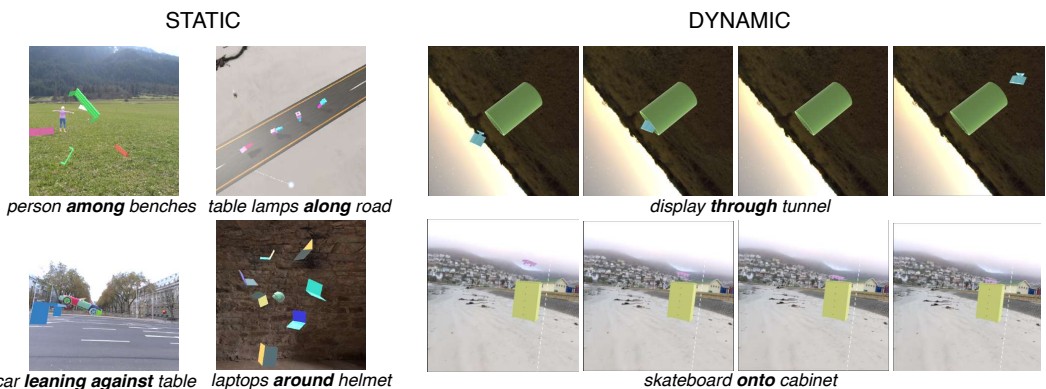

Figure 1: Some examples of Spatial-STUPD, which contains 30 spatial relations. These relations can be divided into two categories - static (involving no motion) and dynamic (involving relative motion between the subject and object)

### 3.4 STATISTICS OF THE STUPD DATASET

#### 3.4.1 SPATIAL-STUPD.

Our primary goal, through this dataset, is to create a well balanced dataset, with a wide and balanced variety of *subject-object* interactions. Firstly, each spatial and temporal relation has 5,000 datapoints each. As mentioned above, we constrain the interaction of supercategories to emulate real-world physical constraints and ensure similarity of size of *subject* and *object*. During dataset generation,

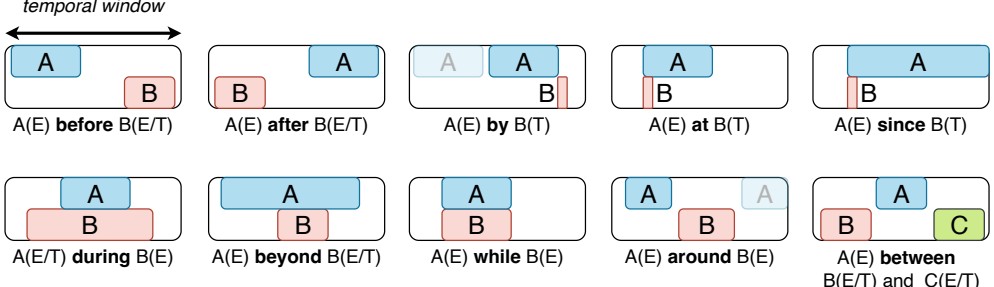

Figure 2: We propose 10 temporal relations representing interactions between different events or time points within a specified temporal window of $W$ frames. Different temporal prepositions are used in specific contexts in English. For each relation, A, B, and/or C can be an event(E), time point(T) or either event or a time point(E/T). Each temporal relation can have multiple types of event/time point interactions. The translucent shade of certain events in the figure represents the possible variation in the point of occurrence.

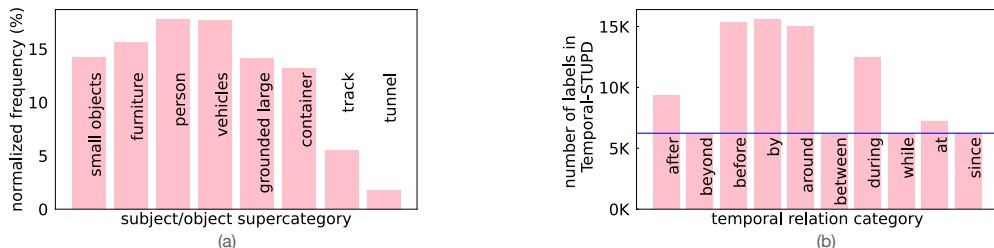

Figure 3: Dataset statistics. (a) The occurrence of prefab categories is roughly consistent throughout the dataset. (b) The blue line represents the minimum number of temporal relation occurrence. A single temporal interaction can have multiple temporal relation predicates associated.

we adjust the number of examples generated for each subject/object supercategory pair based on the total number of object categories interacting, so that individual category occurrences are more or less equal throughout the dataset. In Figure 3(a), we include the distribution of all supercategory occurrences in the STUPD dataset (including both subjects as well as objects). The frequencies are normalized by the number of prefab categories associated with each supercategory and presented as a fraction (percent). As can be seen, the normalized distribution of the majority of supercategories is more or less similar. A couple of observations are as follows.

1. Supercategories 'track' and 'tunnel' have lower frequencies because of their association with only a small number of spatial relations (such as "(movement) **along** track" and "subject (passing) ***through*** a tunnel".

2. It can be seen that the frequency of 'small objects' is slightly lower than others. This is a conscious design choice, because of the size mismatch between other supercategories, having much larger sizes (such as buildings, vehicles, or furniture). We, however, try to maintain balance by including appropriate numbers of interactions between the larger objects within the small objects supercategory and other supercategories.

### 3.4.2 TEMPORAL-STUPD.

Since Events/Time Points are randomly sampled from Spatial-STUPD, the distribution of Events/Time Points is similar to that in Figure 3(a). In Figure 3(b), we illustrate the occurrence of different supercategories across the 50,000 data points. Each predicate has atleast 5,000 occurrences. However, because of the overlap between many temporal relations, many temporal predicates occur more frequently in the dataset. For instance, "before" is a subset of "by", and hence "by" occurs whenever "before" occurs, but not necessarily vice versa. Similary, "while" is a subset of "during" (related to two events occuring simultaneuously) and "since" is a subset of "at" (related to an event occurring at a particular time instance).

# 4 BASELINES

## 4.1 SPATIAL-STUPD BASELINES

In this subsection, we aim to demonstrate that STUPD is an effective pretraining dataset for real world visual reasoning tasks. Ideally, visual reasoning models first pretrained on STUPD, and then transfered to real world datasets, should results in an increase in performance. To demonstrate the effect of STUPD on real world visual reasoning tasks, we choose two real-world visual reasoning datasets - the SpatialSense Dataset (Yang et al., 2019) (to demonstrate performance on static spatial relations) and ImageNet-VidVRD (Shang et al., 2017) (to demonstrate performance on dynamic spatial relations).

### 4.1.1 SELECTION OF BASELINE MODELS

We choose six baselines to evaluate our dataset, inspired by the baselining approach following in (Yang et al., 2019). These models include two simple models (*Language-based model*, which only takes verbal phrases as input, and *Coordinate-based model*, which only takes the coordinates of objects as input) and four deep-learning based model (*DRNet*(Dai et al., 2017), *VIPCNN* (Yikang et al.), *PPRFCN* (Zhang et al., 2017b), and *VTransE* (Zhang et al., 2017a)). While the aforementioned deep-learning based models were specifically designed for visual relationship reasoning, the two simple models were chosen to highlight different aspects of Spatial-STUPD as well as other datasets. For example, the *Language-based* model (which takes the subject and object phrases from a relation triplet, and predicts the predicate) highlights the statistical bias related to subject/object distribution, as is explained in detail in the following subsection. On the other hand, the *Coordinate-based* model (which takes relative spatial coordinates of the subject and object, as well as their bounding box coordinates, and predicts the predicate) highlights the role of coordinate as well as bounding box information and bounding box, while being isolated from any visual features. Hence, through the selection of the various baselines, the role of various components of the dataset can be individually understood.

Additionally, a random baseline is also presented. The models are evaluated on a single label predicate classification task (unlike various previous approaches where the task is a binary classification task to evaluate if the relation triplet, when given as input, holds true). The architecture of the models are adjusted according to the task and dataset used. Further details can be found in appendix A.8. It should be noted that since the architecture of various models has been slightly adjusted to fit the task as well as training dataset, we refer to a model *X* as '*X-based*', to differentiate between the original proposed architecture and the model used in this paper.

### 4.1.2 MODEL PERFORMANCE ON SPATIAL-STUPD

Firstly, to validate the sanity of the STUPD dataset from a model training perspective, we train the baseline models on only the Spatial-STUPD dataset, and present the results in Table2. We note the suboptimal accuracy on the *language-based* model. This is infact, a positive outcome. Predicting the predicate based on only the subject and object category information represents imbalance and/or bias within the dataset. A well-balanced dataset should produce low accuracy on this task, as is seen in the accuracy results. Next, we observe the best performance on Spatial-STUPD is achieved through the *VTranseE-based* model, followed by the *coordinate-based* model, which is a relatively simple model. This demonstrates the higher importance of spatial coordinate/bounding box information over visual features. We also observe the higher performance on dynamic predicates in comparison to static predicates, with the exception of the *DRNet-based* model. This indicates that dynamic data is loaded with more information for spatial reasoning, hence establishing the need for datasets with dynamic information. On the other hand, *DRNet-based* (Dai et al., 2017) model outperforms all models on static data. This special suitability towards images rather than videos may be because of architectural design choices such as the isolation of bounding box feature maps.

### 4.1.3 COMPARISON BETWEEN DIFFERENT PRETRAINING DATASETS

We propose STUPD primarily as an effective pretraining dataset before transfering on real-world dataset. To demonstrate the effect of pretraining a model on STUPD , we compare the results of pretraining on various datasets. For each of the two real-world datasets, we compare the performance with two pretraining datasets – *ImageNet* dataset (Deng et al., 2009b)(for the *SpatialSense* dataset)/*KINETICS-400* dataset (Carreira & Zisserman, 2017)(for the *ImageNet-VidVRD* dataset),

| Model | Overall Accuracy | Static Accuracy | Dynamic Accuracy |
|---|---|---|---|
| Random | 3.34 | 3.34 | 3.34 |
| Language-based | 28.90 | 26.76 | 31.66 |
| Coordinate-based | 75.60 | 72.54 | 78.32 |
| VIPCNN-based | 64.24 | 61.52 | 70.37 |
| PPRFCN-based | 68.19 | 66.41 | 69.47 |
| VTransE-based | **76.58** | 72.22 | **80.39** |
| DRNet-based | 70.32 | **81.35** | 60.70 |

Table 2: Visual reasoning performance trained on all 30 spatial relations in the Spatial-STUPD dataset. The values presented are accuracy metrics in percent.

| (SpatialSense training) | Pretraining dataset | | |
|---|---|---|---|
| **Model** | **no pretraining** | **ImageNet** | **CLEVR** | **Spatial-STUPD** |
| Random | 16.67 | 16.67 | 16.67 | 16.67 |
| Language-based | **43.13** | N/A | 43.04 | 42.91 |
| Coordinate-based | 47.45 | N/A | 47.62 | **49.59** |
| VipCNN-based | 41.17 | 41.94 | 41.11 | **44.28** |
| PPRFCN-based | 44.12 | 42.61 | 42.08 | **44.98** |
| VTransE-based | 49.81 | 49.85 | 46.98 | **50.84** |
| DRNet-based | 51.93 | 52.54 | 52.84 | **54.28** |

Table 3: Effect of Spatial-STUPD pretraining on the SpatialSense (Yang et al., 2019) dataset. The values presented are accuracy metrics in percent.

and the *CLEVR* dataset (Johnson et al., 2017). While the ImageNet/KINETICS-400 dataset serve as general large-scale pretraining datasets for many real-world tasks, the CLEVR dataset is a sythetic dataset with a similar setting as Spatial-STUPD. In general, one of the main purpose of any synthetic dataset is to aid models through additional data proxies for real-world settings, hence serving as effective pretraining options. The results of pretraining on Spatial-STUPD is compared with no pretraining (i.e. direct training on the real world dataset) and other pretraining datasets in Table 3 and Table 4. The details of the training tasks are included in Appendix A.8.

It can be seen that Spatial-STUPD dataset, when used as a pretraining dataset for visual relationship reasoning tasks, improves performance on real-world datasets, especially for deep learning models. On the other hand, CLEVR does not lead to a significant increase in performance in comparison to from-scratch training in most cases. Finally, it can be seen that ImageNet (or KINETICS-400) pretraining infact does not help improve performance in any significant manner. Overall, STUPD is well aligned for various visual relation reasoning tasks, in comparison to other similar synthetic datasets, as well as very large general pretraining datasets like ImageNet/KINETICS.

The fact that ImageNet/KINETICS-400 pretraining does not lead to significant improvement in performance indicates the fact that higher quality visual features do not contribute towards visual relationship reasoning. Effective visual relationship reasoning is a result of other forms of data including bounding box information and relative spatial positioning. This can be confirmed by the performance of Coordinate-only model in the case of ImageNet-VidVRD training, in comparison to any pretraining. It can also be noticed that the jump in accuracy after pretraining is much more pronounced in the case of ImageNet-VidVRD training than SpatialSense training. This indicates the importance of dynamic information for effective visual relationship reasoning.

## 4.2 TEMPORAL-STUPD BASELINES

Extending the pretraining premise from Spatial-STUPD, we formulate a similar task for Temporal-STUPD to demonstrate that models benefit on real-world temporal datasets when pretrained on this dataset. An obvious domain which can benefit directly from pretraining on Temporal-STUPD is visual question answering (VQA) (Manmadhan & Kovoor, 2020), because of the intersection of visually grounded spatial informatin and diverse language-based event descriptions. For the purpose of finetuning a Temporal-STUPD finetuned model on a real world VQA dataset, we choose the *NeXT-QA* dataset (Xiao et al., 2021), which contains structured language-based questions accompanying

| (ImageNet-VidVRD training) | Pretraining dataset | | | |
|---|---|---|---|---|
| Model | no pretraining | KINETICS-400 | CLEVR | Spatial-STUPD |
| Random | 10.00 | 10.00 | 10.00 | 10.00 |
| Language-based | 54.35 | N/A | **55.25** | 54.71 |
| Coordinate-based | 54.49 | N/A | 52.11 | **54.79** |
| VipCNN-based | 50.68 | 50.54 | 58.44 | **86.95** |
| PPRFCN-based | 51.72 | 51.87 | 49.87 | **62.64** |
| VTransE-based | 56.60 | 56.88 | 64.64 | **73.97** |
| DRNet-based | 57.98 | 57.29 | 68.07 | **87.29** |

Table 4: Effect of Spatial-STUPD pretraining on the ImageNet-VidVRD (Shang et al., 2017) dataset. The values presented are accuracy metrics in percent.

| (NeXT-QA training) (Temporal) | Pretraining Dataset | | |
|---|---|---|---|
| Model | no pretraining | KINETICS-400 | Temporal-STUPD |
| Language-based | 50.75 | N/A | **51.37** |
| EVQA-based | 62.69 | 57.52 | **71.42** |
| STVQA-based | 64.24' | 58.76 | **70.52** |

Table 5: Effect of Temporal-STUPD pretraining on the NeXT-QA (Xiao et al., 2021) dataset. The values presented are balanced accuracy metrics, which represent the average of class-wise recall values, in percent. This metric is chosen because of the unbalanced nature of the two relevant classes in NeXT-QA. For this reason, we also omit the random baseline, since in skewed distributions, random predictions lose their value.

videos of everyday events. In Xiao et al. (2021), the authors point out that disambiguation between even simple temporal relations like before/after is difficult for models (both language-only, and vision-language based). We slightly modify the traditional VQA grounding task in order to match the structure of NeXT-QA and Temporal-STUPD. *NeXT-QA* presents language annotations in the form of multiple choice natural-language questions. Corresponding to temporal prepositions, NeXT-QA presents three types of visual questions – An event occuring before another event, an event occuring after another event, and an event occuring at a specific moment in the video. To match the triplet format Temporal-STUPD (*<Event/TimePoint A, relation, Event/TimePoint B>*), we split the question-answer pair string by the relational word – giving us a viable proxy to Events/TimePoints A and B. This temporal relation triplet extraction method is described through examples in the Appendix A.5.1.

### 4.2.1 MODEL PERFORMANCE ON TEMPORAL-STUPD

Table 5 presents the results of pretraining experiments on Temporal-STUPD. Similar to the Spatial-STUPD, we observe sub-optimal performance in the language-based model, and only marginal improvement as a result of pretraining. This is expected, since a language-only model only highlights the statistical nature of However, when language and spatio-temporal fatures are combined, we observe significant improvements in the performance of the models when pretrained on Temporal-STUPD, in comparison to when models are trained on NeXT-QA from scratch. On the other hand, we observe that models, in-fact, perform worse than a model trained from scratch, when pretrained on KINETICS-400. This reinforces two key observations – rich real-world spatial information is not sufficient (or helpful) in effective temporal relation reasoning, and that pretraining on a diverse set of rich temporal relation dataset (like Temporal-STUPD) boosts real-world training significantly.

## 5 LIMITATIONS AND FUTURE WORK

The STUPD dataset was designed with simplicity in mind. A prepositional word can have multiple senses, sometimes with subtle differences in meaning or usage in different contexts. In the case of spatial relations, we restrict context of usage by limiting subjects and objects to physical objects, thus allowing us to group different senses into a single preposition. Further works may focus on creating visual datasets to disambiguate between the subtle meanings of different senses of a preposition. Another dataset design choice was to limit the types of objects to at most 2 types (categories) per image, for simplicity. However, this somewhat limits with number of potential prepositions

included, as some comparative prepositions require 3 types of objects in order to be depicted properly. An example is *as far as*, which depicts a comparison between two distances. This cannot be represented by a scene with interactions between only two objects.

Finally, while 3D information is readily available in STUPD due to its synthetic nature, this was not utilized in this paper, primarily in order to compare the results with previous works. Future works may examine whether and how 3D information may help with certain reasoning tasks.

## 6 CONCLUSION

Static representations such as image based datasets are not sufficient for machine learning systems to fully understand spatial relations well. Spatial relations have many subtle characteristics such as relative movement, velocity, direction, orientation, which can only be fully justified through flexible dynamic representations such as synthetic based videos. In this paper, we introduced a novel dataset which aims to cover the subtle differences between different spatial relations through simple object interactions. Through various experiments, it is evident that the dynamic nature of senses helps model identify relations better. Our studies also demonstrate the nature of spatio-temporal learning in 3D deep learning models. It is observed that models initially rely more on spatial cues, but slowly learn about temporal cues as well, and the combination of spatio-temporal cues results in higher accuracy.

Although this dataset consists of simple object interactions, we hope that it can be used to make models understand more complex scene structures, such as nuanced contexts of preposition use in the English language, or for understanding the underlying dynamics of actions better in various action recognition tasks.

## 7 ETHICS STATEMENT

In the course of conducting our research and developing the dataset for this study, we were committed to upholding ethical standards and ensuring the fair representation of individuals from diverse backgrounds. Our ethical considerations and actions are outlined as follows:

1. **Equitable Representation of Races:** We acknowledge the importance of addressing potential biases in data collection and dataset construction. To mitigate racial biases and ensure fairness, we took deliberate steps to create a dataset where all races are equally represented.
2. **Equitable Representation of Ethnicities:** Recognizing the significance of inclusivity, we made concerted efforts to equally represent various ethnicities within our dataset. We understand that ethnicity encompasses a wide range of cultural and regional identities, and our dataset was designed to reflect this diversity.
3. **Equitable Representation of Age Groups:** Age diversity is a fundamental aspect of our dataset construction. We recognized the importance of capturing the experiences and perspectives of individuals across different age groups.

## 8 REPRODUCIBILITY STATEMENT

We are committed to ensuring the reproducibility and transparency of our research. In accordance with the guidelines set forth by ICLR 2024, we provide detailed information to facilitate the replication of our experiments and results.

1. **Code Availability:** All code used for our experiments is available.
2. **Data Availability:** Any publicly accessible datasets used in our research are specified in the paper, along with their sources and access information.
3. **Experimental Details:** We have documented the specific details of our experiments, including hyper-parameters, model architectures, and pre-processing steps, to enable others to replicate our results.

We are dedicated to supporting the scientific community in replicating and building upon our work. We welcome feedback and collaboration to ensure the robustness and reliability of our research findings.

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
