| Term | Definition |
|---|---|
| *relation* | any interaction between two physical objects or time points. |
| *predicate* | another term for *relation*, in the context of a relation triplet. |
| *preposition* | words that depict different kinds of relations. |
| *prepositional sense* | a particular meaning of a prepositional word, especially in a specific context. the same prepositional word can have different meanings in different contexts. |
| *spatial preposition* | a preposition describing the relation between two physical objects. |
| *static preposition* | a spatial preposition that involves no motion, i.e. the object(s) involved are stationary |
| *dynamic preposition* | a spatial preposition that involves motion, i.e. the object(s) involved are non-stationary |
| *temporal preposition* | a preposition describing the relation between two time points or time periods. |

Table 6: Glossary for the various terms used in this paper.

# A   APPENDIX

## A.1   GLOSSARY OF VARIOUS TERMS USED IN THIS PAPER

Many terms have been used in this paper, which may be related, and hence confusing to the reader. In Table 6, we clearly define a few key words related to our work.

## A.2   PREDICATE VOCABULARY

We use The Prepositions Project (TPP) (Litkowski & Hargraves, 2021b), as a starting point for selecting the prepositions for the purpose of visual reasoning representation in STUPD. The process we implemented to shortlist our list of spatial and temporal prepositions is outlined here.

1. All spatial and temporal prepositions were ranked by the frequency of occurrence in standard text corpora, which acts as a proxy to an indicator of word usage in the English Language.

2. We set a threshold to the frequency of occurrence, and discard all prepositions with a frequency smaller than the threshold since these prepositions are either archaic or have a more frequently used synonym.

3. Then we discard certain prepositional senses based on rules as follows.

   (a) The preposition should not represent a negative sense (e.g. the absence of an object), since they cannot be depicted visually.

   (b) The preposition should not represent interaction between three or more object categories, since this represents a collection of multiple spatial relations, including pairwise relation between objects. This would increase the complexity of our dataset.

   (c) The preposition should not be context-specific (i.e. related to usage with respect to only selected types of objects), since these prepositions create bias within the dataset.

4. One goal of the STUPD dataset was to represent generalizable visual relations, that do not depend on any metric scales of distance (such as kilometers, meters, etc.) and time (seconds, minutes, etc.). For example, the temporal sense "*under*" is used to represent an event that lasts for less than a specific amount of time (e.g. The noodles were ready in *under* two minutes). Hence we discard prepositional senses that require comparison with specific metrics.

5. Many of the remaining prepositions are similar in meaning and can be grouped together (e.g. *("on", "on top of", "upon") or ("below", "beneath", "under", "underneath")*. Hence, we group all prepositions into categories having mutually exclusive definitions,

During the filtering process, we make one specific exemption, where we chose to discard two spatial relation categories - "*to the left of*" and "*to the right of*". This is because of the perspective problem presented by these prepositions - the viewer's (camera's) perspective of these relations can be the

opposite of the *subject*'s perspective, and also different for the *object*'s perspective. This creates ambiguity in the dataset, which is harmful to visual reasoning detection tasks.

This filtering process led to a finalised shortlist of 40 distinct prepositional senses, consisting of 30 spatial prepositional senses and 10 temporal prepositional senses. The 30 spatial prepositional senses are derived from 26 different spatial prepositions, and consist of 14 static spatial relations and 16 dynamic spatial relations. All 40 prepositional senses have distinct meanings and contexts of usage. The list of all prepositions proposed in STUPD, along with their specific definitions/contexts are presented in Table 7.

### A.3 OVERVIEW OF 3D PREFABS SELECTION

The primary goal of STUPD is to create an effective pre-training dataset for real-world visual reasoning tasks. As such, we first filter out frequently occurring objects in three real-world visual reasoning datasets (the Visual Genome dataset, (Krishna et al., 2016), the SpatialSenses dataset (Yang et al., 2019) and the ImageNet VidVRD dataset (Shang et al., 2017) and categorize all object categories into clusters based on similarity into supercategories (e.g., "car", "motorcycle" and "bicycle" can be categorized into *vehicles*).

Based on these clusters, we select related object prefabs from the ShapeNet dataset (Chang et al., 2015), including but not limited to the categories identified from the real-world visual reasoning datasets mentioned above (continuing the aforementioned example, we select other *vehicle* prefabs as well, such as "bus", "boat" and "train"). This is done to increase statistical variation in the STUPD dataset. Some other 3D prefabs, such as *person*, *tunnel*, and *track (road)* were added to the STUPD repository from various community platforms that provide free 3D prefabs for open use. All prefabs are manually resized to maintain relative size (e.g. a soda can is smaller in size than a table lamp). The ranges of relative size for objects are chosen based on the average size of the object in the real world.

We also try to address certain ethical concerns in our object collection. Specifically, to avoid bias in visual relation detection tasks, we curate 6 instances of *person*, with an equal number of male and female characters, with varying skin color, age, height, and clothing. Figure 4 depicts all the prefabs that were used for the generation of the STUPD dataset. Some statistics of the race (skin color), age and gender are depicted in Figure 5.

We allow only selective pairs of supercategory interactions for each spatial relation, to maximize realism in the dataset. For example, consider the relation predicate <*above*>. It is unlikely to come across an image example of <*building above person*>, since a building cannot hover in the air. Additionally, we also filter out supercategory pairs with a large difference of size. For example, we avoid interaction between a train and a coffee mug, since the size difference is so large, that both cannot simultaneously fit into the same image without compromising the visibility of at least one of these objects. Thus our supercategory pair filtering process involves filtering out all infeasible supercategory <*subject, object*> combinations based on physical constraints and differences of size.

Further, to induce realism in the STUPD dataset, we curated 10 background images containing a variety of sceneries, such as blue sky, ocean, grass fields, desert, cityscape, etc. The backgrounds are carefully chosen so there are no visual distractions, such as objects that might confuse computer vision models. Additionally, we include variation in lighting direction and intensity, thus allowing us to create a realistic shadowing effect as well.

Our approach gives the STUPD dataset various advantages over previous synthetically generated visual reasoning datasets, such as CLEVR (Johnson et al., 2017) and Rel3D (Goyal et al., 2020). Since we use real-world objects and backgrounds in our synthetic dataset, along with realistic physical interactions, rather than simple objects such as cubes or spheres, our dataset can transfer knowledge into real-world visual reasoning tasks much more efficiently. Additionally, while we filter out unrealistic object interactions, we simulate interaction between many object categories that are rarely found in real-world datasets such as Visual Genome (Krishna et al., 2016), and SpatialSenses (Yang et al., 2019) (for example <*car on car*>). The diverse range of object interactions in STUPD will allow visual reasoning models to learn spatial relations by learning only the interaction between objects, and not the type of objects (which is a kind of bias inherent in many real-world visual reasoning datasets). There are many potential benefits of using the STUPD dataset in real-world

| Prepositional Type | Preposition | Definition |
|---|---|---|
| Spatial - static | *above* | at a higher level than |
| | *against* | in physical contact with (something), so as to be supported by |
| | *all over* | everywhere |
| | *along* | extending in a more or less horizontal line on |
| | *among* | situated more or less centrally in relation to (several other things) |
| | *around* | on every side of, so as to encircle (someone or something) |
| | *behind* | to the far side of (something), typically so as to be hidden by it |
| | *below* | at a lower level than |
| | *beside* | at the side of (next to) |
| | *between* | at, in, or across the space separating (two objects) |
| | *in front of* | in a position just ahead or at the front part of someone or something else |
| | *inside* | situated with the boundaries of (a container) |
| | *on* | physically in contact with and supported by (a surface) |
| | *outside* | situated beyond the boundaries or confines of (a container) |
| Spatial - dynamic | *against* | moving in the opposite direction of |
| | *along* | moving in a constant direction on |
| | *around* | so as to pass (a place or object) in a curved or approximately circular route. |
| | *by* | so as to go past |
| | *down* | movement from a higher to a lower point |
| | *from* | indicating the point in space at which a motion starts in the direction away |
| | *into* | expressing movement with the result that something becomes surrounded by (a container) |
| | *into* | expressing movement (collision) with the result that something makes physical contact with something else |
| | *off* | moving away from |
| | *onto* | moving to a location on the surface of |
| | *out of* | movement from within the boundaries of (a container) |
| | *over* | expressing passage or trajectory directly upwards from |
| | *through* | moving in one side and out of the other side of |
| | *towards* | moving in the direction of |
| | *up* | movement from a lower to a higher point |
| | *with* | movement in the same direction as |
| Temporal | *after* | in the time following (an event or another period of time) |
| | *around* | occurring slightly before or after a given point of time |
| | *at* | expressing the exact time when an event takes place |
| | *before* | in advance of the time when |
| | *beyond* | happening or continuing after (a specified time or event) |
| | *between* | in the period separating (two events or points in time) |
| | *by* | indicating a deadline or the end of a particular time period |
| | *during* | at a particular point in the course of (an event) |
| | *while* | coinciding at the same time of; meanwhile |
| | *since* | in the intervening period between (the time mentioned) and the time under consideration, typically the present. |

Table 7: List of all prepositional senses used in STUPD and their definitions. In total, we derive 40 prepositional senses from 34 distinct prepositions, of which there are 30 spatial prepositional senses (derived from 26 distinct prepositions) and 10 temporal prepositional senses.

settings. A simple example is *<person (moving) down>*. Because not many examples of this event may be found in real-world visual reasoning tasks, it can cause visual reasoning models to fail to detect an event such as a *<person falling down from a building>* in real-world settings. Our dataset aims to fill this knowledge gap for visual reasoning models.

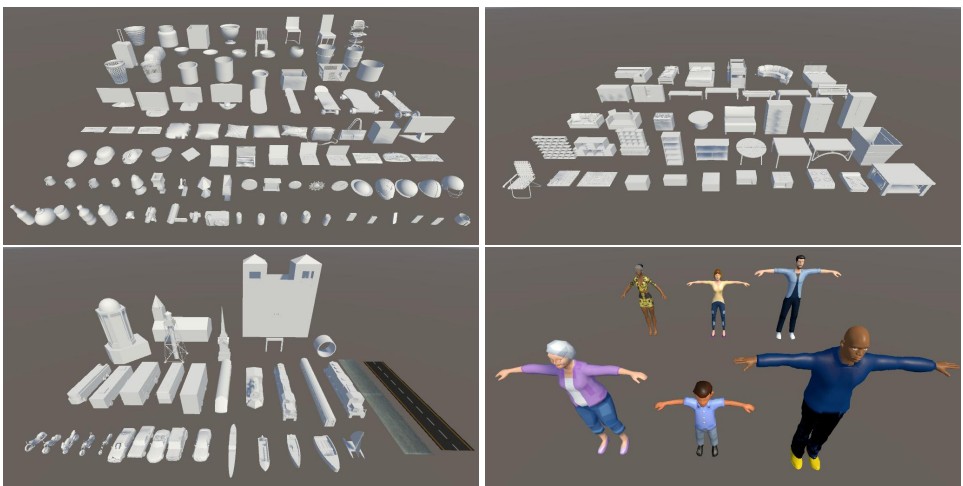

Figure 4: Overview of 3D prefabs used. We curate a total of 183 prefabs across 45 categories and 8 supercategories. We also try to have a balanced set of *person* prefabs to address certain ethical concerns (bottom right). (Also see Figure 5)

### A.4 SPATIAL-STUPD DATASET

We present some other examples of spatial relations in this section. Refer to Figure 6 for examples of static spatial relations and Figure 7 for examples of dynamic spatial relations.

### A.5 TEMPORAL-STUPD DATASET

In the Temporal-STUPD dataset, the events vary from $f' = 15$ to $f' = 45$ frames, and are represented uniformly. In the video, only the frames corresponding to the occurrence of the event/time point are visually illuminated, and the remaining sections of the video are blacked out. This approach gives certain advantages to us.

1. Since all temporal relations are independent of the nature of events themselves (for example, in *<Event A, after, Event B>*, Event A and B can be any spatial relation occurrence), our method segregates the events and time points into different video streams, allowing temporal reasoning models to effectively understand this independence.

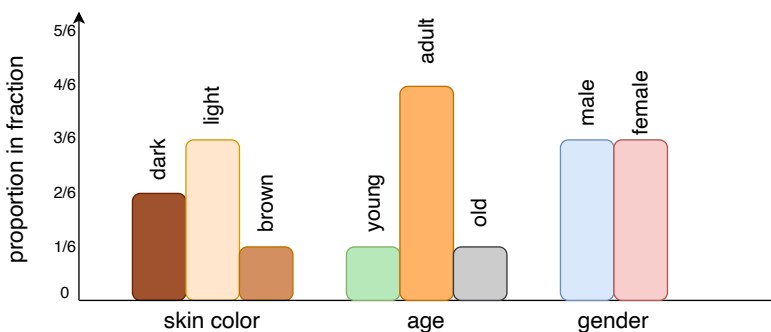

Figure 5: Ethnic distribution statistics for the *person* prefab.

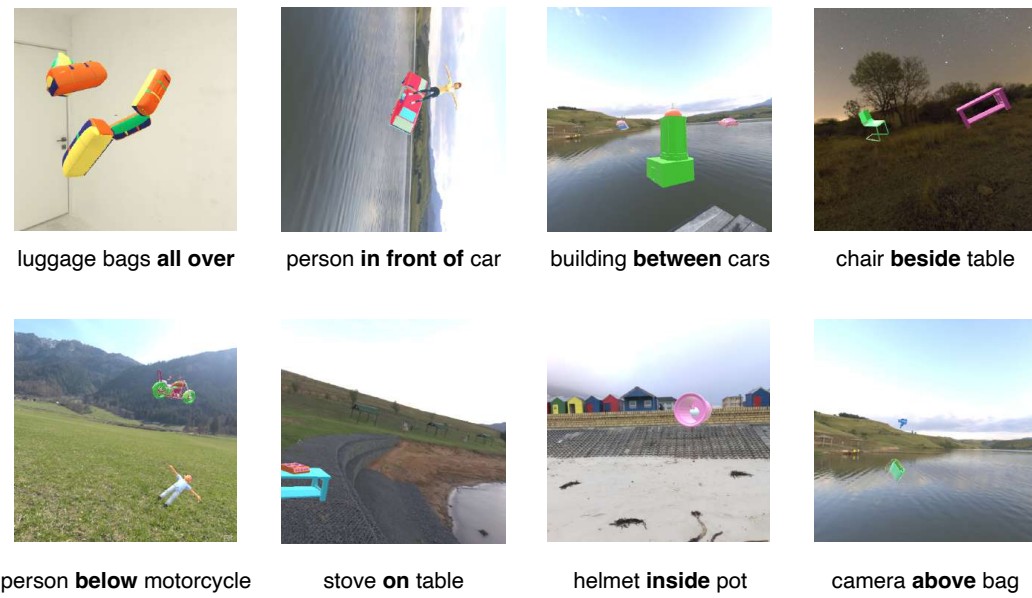

| | | | |
|---|---|---|---|
| luggage bags **all over** | person **in front of** car | building **between** cars | chair **beside** table |
| person **below** motorcycle | stove **on** table | helmet **inside** pot | camera **above** bag |

Figure 6: Some other static spatial relations in STUPD

2. Our approach allows us to scale the temporal interaction to >2 events/time points. For example, consider the temporal predicate <*between*>, which involves interaction between 3 events/time points (<*Event A occurs between Event B and Event C*>). Hence we provide annotations of each temporal relation in the form of information about the temporal relation category, and information about *Event A*, *Event B*, and *Event C* (where *Event C* field can be empty), such as which spatial event it corresponds to, along with the frame numbers corresponding to the start and end of any event/time period.

An event of $f'$ frames ($f' \epsilon [15, 45]$) can be represented by a static event (where the same image is spread across $f'$ frames, representing the occurrence of a static event for a particular length of time) or a dynamic event (where the original $f = 30$ frames are interpolated by uniform sampling (if $f' \lower f$) or spherical interpolation (if $f' \lower f$). Spherical interpolation ensures that the corresponding video is smooth even if the frame rate for a dynamic spatial interaction is reduced.

An overview of temporal relations in STUPD is presented in Figure 8.

### A.5.1 PRETRAINING EXPERIMENT DESIGN

We use the NeXT-QA dataset as an example of a real-world dataset, and perform experiments to see if pretraining on Temporal-STUPD improves real-world temporal relation reasoning. However, most standard VQA datasets have a question-answer pair format. In order to pretrain models on Temporal-STUPD, followed by finetuning on NeXT-QA, we introduce a temporal relation triplet extraction heuristic, that utilizes the linguistic structure of sentences involving the words 'before' and 'after', which may be described as <*Question asking about event A*> before/after <*description of event B*>? <*Description of event A*>. Using this structure, we convert the natural-language question-pair into structured relation triplets as proposed in this paper.

For example, consider the question-answer pair from NeXT-QA: *"What happens **after** the lady pushes the girl? The girl slides down."* A simple sentence manipulation logic based around the word "after" yields the temporal relation triplet <*the girl slides down, after, the lady pushes the girl*>. In the case of NeXT-QA, we notice that while question/answers involving *A preceding B*, and *B preceding A* involve just two prepositional relations (before and after), the examples involving *A occuring at a specific moment* containa a diverse range of natural language structures, not necessarily involving any temporal relations. Hence, for structural integrity, we only evaluate pretraining effects of Temporal-STUPD for the questions involving *A before B*, and *A after B*. An analysis of the triplet

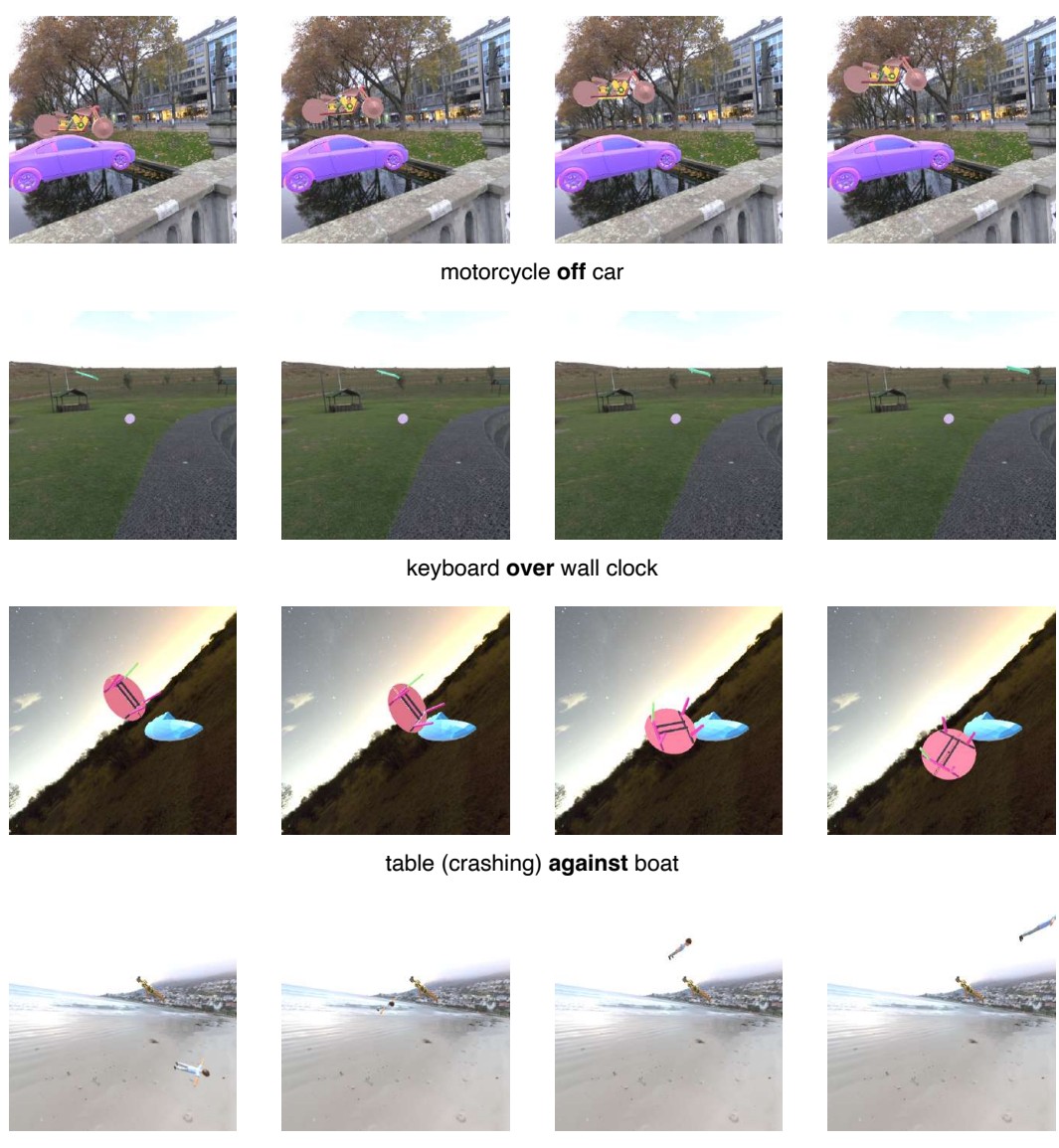

Figure 7: Some other dynamic spatial relations in STUPD. Note that some predicates may have multiple senses (definitions) in different context. We represent all 30 spatial senses with a unique label in the STUPD dataset.

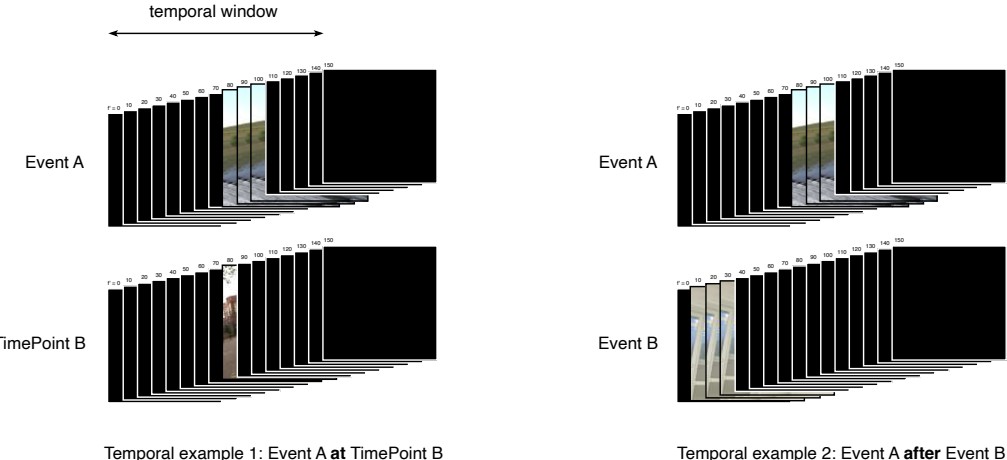

Figure 8: Examples of temporal relations in STUPD. Each temporal relation can be a relation in time between 2 or more events or time points. Each event (spanning over multiple frames in the temporal window) is represented through either a static or dynamic spatial relation, while each time point (a single frame in the temporal window) is represented by a static spatial relation.

extraction heuristic reveals that most of language-based context is preserved in the triplet across all temporal examples in the NeXT-QA dataset.

We perform classification between only 'before' and 'after' in NeXT-QA. To match the video format in NeXT-QA, we combine the video pairs in Temporal-STUPD by overlaying the videos on each other, resulting in a video with a sequence of distinct events.

### A.6 CHOICE OF REAL-WORLD DATASET

To demonstrate that Spatial-STUPD is an effective pretraining dataset for spatial reasoning tasks, we evaluate fine-tuning of various models pretrained on STUPD, on ImageNet-VidVRD (for dynamic spatial relations) and SpatialSense (for static spatial relations). While other datasets, especially for static spatial relations, could also be included in our analysis. Unfortunately, at the time of writing this paper, Visual Genome (Krishna et al., 2016) and VRD (Lu et al., 2016) datasets were not accessible due to server issues. Therefore, we were unable to retrieve the necessary data and incorporate them into our research. We acknowledge that these datasets could have potentially provided valuable insights and comparisons for our study. However, the unavailability of these datasets prevented us from including them in our analysis.

In the case of Temporal-STUPD, we evaluate fine-tuning of various model on the NeXT-QA dataset (Xiao et al., 2021). This dataset has certain advantages over other VQA dataset. It covers certain simple temporal relations (before/after/during), unlike most VQA datasets that focus on spatial relations only, or do not provide sophisticated annotations for the temporal relations involved. Secondly, the structured annotation format makes it easy to convert natural-language question-answer pairs into triplets, that directly match the format of annotations provided in STUPD.

### A.7 DATASET REPRODUCIBILITY

Throughout the dataset generation process, many parameters were assumed. To provide greater flexibility to the users of this dataset, we release all dataset generation scripts, environments, prefabs, parameter value sheets, and documentation to allow users to recreate or adjust values (and/or logic of the spatial/temporal dataset generation). Thus, while we release 150,000 spatial relations and 50,000 temporal relations, in theory, a lot more images can be generated as seen fit by users. We observed that generating the dataset in small batches with different seed values in UNITY results in a diverse dataset.

## A.8 TRAINING DETAILS

### A.8.1 TRAINING SPLITS FOR THE STUPD DATASET

We intend this dataset to be randomly split into training, validation and/or testing splits, and hence do not provide any explicit labels for this. This design choice arises due to many reasons. First, since this dataset aims towards being a pretraining dataset, performance metrics on STUPD should not be strictly optimized, and only used as a rubric to understand a models understanding of spatial/temporal relations. Secondly, a random split is representative of the balanced distribution of STUPD. Due to this reason, we also do not report the accuracy of predicate classification on the STUPD dataset during pretraining stages.

### A.8.2 BASELINE MODEL DESIGN PROCESS

**Spatial-STUPD baselines**  In this paper, we design the task to be a single-label predicate classification task. Various visual relation detection models (including the models used in this paper for baselining) showcase model performance through a binary classification task, where the information of the subject and object, as well as the predicate is fed as input to the models, and the task is to evaluate if the relationship triplet holds true. However, a binary classification approach does not clearly indicate the performance of models, because the answer being only one of two options. In this paper, our approach is to only feed models with information about subject and objects, and predict the predicate as a single-label classification task. While the accuracy achieved through this technique is much lower than a binary classification task, the granularity allows us to understand the differences between different models and tasks clearly. To adjust for the modified task, the architectures of the various models has been adjusted such that no information of the predicate is fed as input, and the final layer is replaced with a single-label classifier.

It should also be noted that the models used for baselining in this paper are primarily designed for image data. Hence, in order to support training over dynamic data, such as Imagenet-VIDVRD (Shang et al., 2017), we modify the architectures by replacing 2D convolutional backbones with ResNet-based 3D convolutional layers (Tran et al., 2018), optionally pretrained over the KINETICS-400 dataset (Carreira & Zisserman, 2017) (as opposed to the ImageNet dataset in the traditional 2D convolutional network). All the code is published in a public repository and can be found here.

For pretraining, we select only a part of the STUPD dataset. For all pretraining tasks, we select predicate categories that are shared by both the real-world dataset and the STUPD dataset. This leads us with a 6-way single-label classification task for the SpatialSense dataset, and a 10-way single label classification task for the ImageNet-VidVRD dataset. A similar strategy is followed for the CLEVR dataset. In the case of ImageNet dataset, we only pretrain the convolutional backbone of the model. For a fair comparison, we train all models with the same hyperparameters and number of epochs of training. All pretraining tasks (except ImageNet pretraining) are carried out for 2 cycles, while finetuning on the real-world task is carried out for 5 cycles. The ImageNet pretraining is carried out for 20 cycles.

**Temporal-STUPD baselines**  The corresponding predicate classification task for Temporal-STUPD involves a binary classification problem (corresponding to the classes *'before'* and *'after'*), because of certain limitations of the concerned real world dataset (NeXT-QA) (see A.5.1). We select three models to demonstrate the effect of pretraining on Temporal-STUPD, namely the language-based model (similar to Spatial-STUPD), the EVQA-based model (Antol et al., 2015), and STVQA-based model Jang et al. (2019). The architectures of EVQA and STVQA models have been slightly modified to adapt to the relation predicate classification task. The language component, which is fed as input to all these models involves parsing both Events/TimePoints A and B (in the temporal relation triplet) through a word embedding model, and concatenated into one single vector. Then this vector is parsed through an LSTM model (Yu et al., 2019). In the case of EVQA and STVQA models, the input video features (stack of images) is parsed through a Resnet18-3D model, and the fully connected layer activations are dot-multiplied with language embeddings, before being fed into the final fully-connected classification layer. Except the KINEMATIC-400 pretrained model, all other models take 10 frames from sampled uniformly from the video (ie, 10 temporally equidistant frames). For the backbone pretrained on KINEMATIC-400, we only use 3 frames, because KINEMATIC-400 pretrained models sued 3 input channels only.

### A.9 DATASHEETS FOR DATASET

In this section, we provide the datasheets for the STUPD dataset.

#### A.9.1 MOTIVATION

The questions in this subsection are primarily intended to encourage dataset creators to clearly articulate their reasons for creating the dataset and to promote transparency about funding interests.

1. **For what purpose was the dataset created?** The STUPD dataset is a large-scale synthetic dataset primarily designed to act as an efficient pretraining dataset for visual reasoning tasks in real-world settings. It also is a balanced dataset representing a wide variety of spatial and temporal relations, all with distinct definitions.

2. **Who created the dataset (e.g., which team, research group) and on behalf of which entity (e.g., company, institution, organization)?** Created by Center for Frontier AI Research (CFAR), A*STAR, Singapore.

3. **Who funded the creation of the dataset?** This project was funded by A*STAR internal funding.

#### A.9.2 COMPOSITION

The questions in this section are intended to provide dataset consumers with the information they need to make informed decisions about using the dataset for their chosen tasks. Some of the questions are designed to elicit information about compliance with the EU's General Data Protection Regulation (GDPR) or comparable regulations in other jurisdictions.

1. **What do the instances that comprise the dataset represent (e.g.,documents, photos, people, countries)?** RGB images of standard dimensions 512x512, depicting interaction between different types of 3D objects.

2. **How many instances are there in total (of each type, if appropriate)?** There are 200K instances in total - 150K instances for spatial relations and 50K instances for spatial temporal relations. The spatial relation instances are evenly distributed across 30 prepositional relations, each consisting of 5,000 examples each. Out of the 30 relations, there are 14 static relations, which are represented by 1 image per instance, and 16 dynamic relations, represented by 30 images per instance (depicting a video of 30 frames total). Each temporal instance contains one or more streams of videos containing 150 images each. We provide all videos in the form of images. In total STUPD is a collection of 17,920,000 images.

3. **Does the dataset contain all possible instances or is it a sample (not necessarily random) of instances from a larger set?** No, the dataset has been curated (generated) from scratch.

4. **What data does each instance consist of?** Each data instance is in the form of an RGB image.

5. **Is there a label or target associated with each instance?** We provide annotations for each data instance, providing information about the objects in the image(s), their nature (which category and supercategory they belong to), the type of spatial/temporal relation that the image is part of, the 2D bounding boxes of all objects in the image, and the 3D spatial coordinates of the center of the objects in the image.

6. **Is any information missing from individual instances?** No, the images have been provided as generated by Unity through scripts, which we publish for open-source access.

7. **Are relationships between individual instances made explicit?** The annotations make clear which images belong to which data instance.

8. **Are there recommended data splits (e.g., training, development/validation, testing)?** We recommend a random split for training purposes. The goal of STUPD dataset is to act as an effective pretraining dataset.

9. **Are there any errors, sources of noise, or redundancies in the dataset?** There are a few images with minor errors. For example, some objects may be partially or, in extreme

cases, completely out of the field of view. Another source of error may be the overlapping of 3D objects, which may look unnatural if compared with real-world settings. However, in most cases, the core relation represented by the image still holds true, and a combination of information from images and annotations contains all the required information for effective visual relation reasoning.

10. **Is the dataset self-contained, or does it link to or otherwise rely on external resources (e.g., websites, tweets, other datasets)?** Our dataset is self-contained. In fact, we also provide all required scripts and environments for users to regenerate or modify our dataset as they see fit for their own purpose.

11. **Does the dataset contain data that might be considered confidential (e.g., data that is protected by legal privilege or by doctor–patient confidentiality, data that includes the content of individuals' non-public communications)?** No.

12. **Does the dataset contain data that, if viewed directly, might be offensive, insulting, threatening, or might otherwise cause anxiety?** No.

### A.9.3 COLLECTION PROCESS

In addition to the goals outlined in the previous section, the questions in this section are designed to elicit information that may help researchers and practitioners to create alternative datasets with similar characteristics. We only address relevant questions in this section.

1. **How was the data associated with each instance acquired?** The data was generated by Unity3D through a custom scripting process.

### A.9.4 PREPROCESSING/CLEANING/LABELING

The questions in this section are intended to provide dataset consumers with the information they need to determine whether the "raw" data has been processed in ways that are compatible with their chosen tasks.

1. **Was any preprocessing/cleaning/labeling of the data done (e.g., discretization or bucketing, tokenization, part-of-speech tagging, SIFT feature extraction, removal of instances, processing of missing values)?** We removed redundant information from the meta files generated by Unity during the dataset generation process, and only retained the information necessary for visual reasoning tasks.

2. **Was the "raw" data saved in addition to the preprocessed/cleaned/labeled data (e.g., to support unanticipated future uses)?** Yes, users can request for the raw metadata files generated by Unity by contacting the authors. Additionally, we provide data value sheets that contain all information about parameters that users can use to regenerate the dataset as originally intended.

3. **Is the software that was used to preprocess/clean/label the data available?** Yes, Unity3D is an openly available software. We also release the environment files that will aid users to quickly setup their application to generate data.

### A.9.5 USES

The questions in this section are intended to encourage dataset creators to reflect on the tasks for which the dataset should and should not be used. By explicitly highlighting these tasks, dataset creators can help dataset consumers to make informed decisions, thereby avoiding potential risks or harms.

1. **Has the dataset been used for any tasks already?** No, we introduce this dataset for the first time in this paper.

2. **Is there a repository that links to any or all papers or systems that use the dataset?** Yes, will be provided later.

3. **What (other) tasks could the dataset be used for?** Potentially, this dataset can be used to transfer knowledge from the synthetic domain to the real-world domain, and to learn the dynamics of physics from the interaction of different kinds of objects.

4. **Is there anything about the composition of the dataset or the way it was collected and preprocessed/cleaned/labeled that might impact future uses?** No.

### A.9.6 DISTRIBUTION

Dataset creators should provide answers to these questions prior to distributing the dataset either internally within the entity on behalf of which the dataset was created or externally to third parties.

1. **Will the dataset be distributed to third parties outside of the entity (e.g., company, institution, organization) on behalf of which the dataset was created?** No.

2. **How will the dataset will be distributed (e.g., tarball on website, API, GitHub)?** We will provide a link where users can download the entire dataset from a server. Links will be provided later.

3. **Will the dataset be distributed under a copyright or other intellectual property (IP) license, and/or under applicable terms of use (ToU)?** The dataset will be provided under the **CC BY-NC-SA 4.0** license. The Unity scripts and environment files will be released under the **GNU General Public License 3.0**.

4. **Do any export controls or other regulatory restrictions apply to the dataset or to individual instances?** No.

### A.9.7 MAINTENANCE

The questions in this subsection are intended to encourage dataset creators to plan for dataset maintenance and communicate this plan to dataset consumers.

1. **Who will be supporting/hosting/maintaining the dataset?** The authors will support and maintain the dataset. Access to the dataset via servers will be supported by A*STAR internal funding.

2. **How can the owner/curator/manager of the dataset be contacted (e.g., email address)?** Users can contact the authors with their official email addresses.

3. **Is there an erratum?** (to be updated later)

4. **Will the dataset be updated (e.g., to correct labeling errors, add new instances, delete instances)?** Yes, if required.

5. **Will older versions of the dataset continue to be supported/hosted/maintained?** No, unless there is a clear productive use of an older version.

6. **If others want to extend/augment/build on/contribute to the dataset, is there a mechanism for them to do so?** We provide all resources for researchers to build on top of our dataset, including but not limited to regenerating the dataset, changing the logic of the dataset, adding more objects(prefabs), enhancing the physics engine for interactions, and modifying parameters of the dataset generation process.