# OpenReview forum: "STUPD: A Synthetic Dataset for Spatial and Temporal Relation Reasoning"
_ICLR.cc/2024/Conference — Submitted to ICLR 2024_

### Official Review · Reviewer_NbSm · 2023-10-15

**Soundness:** 3 good
**Presentation:** 3 good
**Contribution:** 2 fair
**Rating:** 5
**Confidence:** 3

**Summary:**

The paper proposes a new synthetic dataset: the Spatial and Temporal Understanding of Prepositions Dataset (STUPD). It is a large-scale video dataset for understanding spatial and temporal relationships. The dataset contains 150K visual depictions consisting of 30 static and dynamic spatial prepositions and 50K visual depictions across 10 temporal relations.
The synthetic dataset helps models perform better in visual relationship detection in real-world settings, verified on 2 real-world datasets: ImageNet-VidVRD and Spatial Senses.

**Strengths:**

The proposed dataset STUPD makes contributions to relation reasoning:
- It covers diverse spatial and temporal relations: 30 spatial prepositions and 10 temporal prepositions
- It elaborates on the spatial relations that intrinsically involve motion
- It is a large-scale dataset

Experiments show that pretraining on STUPD increases performance on real-world visual reasoning tasks.

**Weaknesses:**

1. Details missing about the evaluation of  STUPD pre-training. From the suppl Sec.A.8.2, the SpatialSense/ImageNet-VidVRD experiment only covers 6/10 spatial relations.
    - I have not found details about which relations have been conducted experiments on. ImageNet-VidVRD defines 132 predicates. Some of them are "static"  but not "dynamic" spatial relations. Also, the spatial relations are connected with verbs e.g., swim_behind, and fly_behind. It is not clear howto handle the different definitions of spatial relations during pre-training.
    - There are some uncovered spatial prepositions among the 30 defined spatial relations. It seems that they have been collected but not evaluated. Since STUPD is a synthetic dataset that suffers from huge gaps with real-world images. It is important to verify its effectiveness on real-world tasks.

2. I wonder about the necessity of prosing a new task of temporal relations. Their definition is detailed in Fig.2. Instead of a specific model to reason these temporal relations, it seems that we can directly apply temporal segmentation of each event and then compare the two segmented time stamps. For example, to judge ”A before B”, we first get the temporal segmentation A: $[t_1, t_2]$, B:$[t_3,t_4]$; if $t_2 \textless t_3$, then we say “A before B”.

**Questions:**

See "Weakness".

---

> ### Author Response · Authors · 2023-11-21
> **Clarifications on the paper**
>
> Dear Reviewer, thanks for taking the time to review our paper.
>
> 1. " have not found details about which relations have been conducted experiments on. ImageNet-VidVRD defines 132 predicates. Some of them are "static" but not "dynamic" spatial relations. Also, the spatial relations are connected with verbs e.g., swim_behind, and fly_behind. It is not clear howto handle the different definitions of spatial relations during pre-training."
>
> Thanks for pointing this out. We will add more details to the paper.
> In the ImageNet-VidVRD, we combine all similar relations (eg, behind, swim_behind, crawl_behing, fly_behind). "Swim_behind", for example, is just a subset of behind.
> As to how we handle static vs dynamic relations -- we convert all relations into "dynamic" representations, i.e., the same 3D model handles both dynamic and static relations. Static images are copied n times to create a 3D representation. Our class-wise experiments (in Table 2) suggest that the model can disambiguate between true dynamic 3D representations, and static 3D representations.
>
> 2.  "There are some uncovered spatial prepositions among the 30 defined spatial relations. It seems that they have been collected but not evaluated. Since STUPD is a synthetic dataset that suffers from huge gaps with real-world images. It is important to verify its effectiveness on real-world tasks".
> Unfortunately, there are no real-world visual relation datasets, to our knowledge, that cover the variety of relations that we have collected. This is, thus, a reflection for other researchers to expand their work to cover these relations as well. For current datasets that cover the limited number of spatial relations, it makes sense to only pretrain on the subset of STUPD that overlaps with the datasets. This helps us perform systematic ablations about the specific knowledge of individual relational meanings imparted by STUPD.
> As to whether or not the uncovered relations will be useful for real-life settings, our experiments provide strong arguments in that direction.
> We systematically showed that STUPD-pretrained models perform better on real-world datasets containing relations as we define in our paper. Since, the data designing and generation process is consistent across all prepositions, it is logical to deduce that overall, other prepositional relations in STUPD would also be useful to real-world settings.
>
> 3. "I wonder about the necessity of prosing a new task of temporal relations."
> Actually your argument is correct. An oracle model with time-stamp based inputs is not really useful, since one can differentiate between many relations with simple rules.
> Based on another reviewers comment, we introduced another experiment (involving a VQA dataset), that shows how pretraining on temporal STUPD can also help in real-world settings. In this model, we evaluate whether models can learn temporal relations by analysing the videos, and understanding the nature of the individual events. We included many changes in the paper, all highlighted in the color blue.

---

### Official Review · Reviewer_RbeZ · 2023-10-26

**Soundness:** 1 poor
**Presentation:** 2 fair
**Contribution:** 2 fair
**Rating:** 3
**Confidence:** 5

**Summary:**

Identifying temporal relationships, compared to the spatial relationship counterpart, has less attention in the field of computer vision so far. Thus, the authors are focused on generating a dataset that focuses on both spatial and temporal reasoning. Though the motivation sounds great to the researchers in this community, it is hard to understand why the temporal relationship should designed in the way the author presents. Also, it is hard to find how the temporal dataset could be used for other real-world tasks.

**Strengths:**

The main motivation for the dataset creation, which focuses on both spatial and temporal reasoning, is clear, and the dataset comparison table (Table 1) helps the readers understand the landscape of the field. Also, applying the dataset to two spatial real-world tasks with multiple baseline models also well-represents the effectiveness of this dataset. Lastly, visualization of the dataset helps the reader understand what the dataset looks like.

**Weaknesses:**

Although this dataset (partially) focused on temporal relationships, it is hard to understand why such categorization (in Figure 2) is valid. Also, I failed to find any experiment employing the STUTD dataset to improve the performance on real-world *temporal* relationship tasks. Apart from the main content, this manuscript may violate Sections 2 and 4.1 of the ICLR 2024 author's guidelines.

**Questions:**

[Major]

A. Lack of use of the STUPD dataset as a temporal reasoning pretraining set. Unlike what the authors mentioned at the end of Section 4.2, there is a lot of video-based work that focuses on temporal reasoning. For instance, many datasets [1,2,3,4] exist in the video question-answering domain. It would be better to employ some models that try to resolve the tasks suggested by such datasets.

B. Reason for splitting the temporal relationship into ten categories. Based on what the author said, "before" is a subset of "by," "while" is a subset of "during," and "since" is a subset of "at" (from 1st para of Sec 3.4.2). Also, I don't think "by" (which implies a deadline) is used to describe the timing of two events in general. At least from my end, it is natural to say, "Turn in the assignment by midnight" instead of "Cut off the corners of the bread by the time you apply jam on the bread," for example. The author also mentioned 'redundant representation' in Section 2.1; in this regard, I failed to find any reason for keeping potentially redundant classes. Isn't it more natural to compress the classes into 7 instead of 10? If the author firmly believes a 10-class setting is much more meaningful, then I think it would be better to have an experiment in A but present the result with 7-class pretraining and 10-class pretraining.

C. Clarity. In the #1 callout in Sec 3.4.1, the author pointed out that 'track' and 'tunnel' have fewer relationships than other object types. Why did such a case happen? Is it because of the limitation of the Unity3D platform, or is it because all the relationships came from another dataset?



[Minor]

A. Formatting
- \citet and \citep are different. Please carefully check the ICLR 2024 author's guidelines; It is improperly used in over 80% of the manuscript.
- The author frequently used  "<" and ">" without any escape character. Thereby, those symbols are repeatedly presented as flipped '!' and '?' characters throughout the text.

B. Typo
- 2nd line of 2nd para of Intro (...in space of time" **pre**. Examples of...): I cannot understand what **pre** is for.
- 50,000 uses the middle comma, but 5000 doesn't (always?) throughout the text.


[References]

[1] Jang et al., TGIF-QA: Toward Spatio-Temporal Reasoning in Visual Question Answering, in CVPR 2017.

[2] Mun et al., MarioQA: Answering Questions by Watching Gameplay Videos, in ICCV 2017.

[3] Xiao et al., NExT-QA: Next Phase of Question-Answering to Explaining Temporal Actions, in CVPR 2021.

[4] Li et al., From Representation to Reasoning: Towards both Evidence and Commonsense Reasoning for Video Question-Answering, in CVPR 2022.

---

> ### Author Response · Authors · 2023-11-23
> **Official Comment**
>
> Dear Reviewer, thanks for taking the time to read our paper in depth and provide critical comments. Your reviews helped us improve the quality of the paper significantly. We try to answer your concerns below.
>
> 1. "Lack of use of the STUPD dataset as a temporal reasoning pretraining set.". You are right, and after several discussions, we decided to proceed with a pretraining experiment for Temporal-STUPD as well, which involves a VQA dataset. While most VQA datasets either focus on spatial relations only, or do not provide structured temporal relation annotations, we observe that the NeXT-QA dataset [1] is suitable for our needs. Another candidate -- the TEMPO dataset [2] would have been useful, but unfortunately the dataset has not been uploaded publically in its entirety by the authors. Please see Section 4.2 of the revised paper. Similar to Spatial-STUPD, we observe that Temporal-STUPD too helps in improving real world temporal relation reasoning significantly.
>
> 2. "Reason for splitting the temporal relationship into ten categories [is unclear]". The main goal of STUPD is to not just provide synthetic visual representations of common prepositional relations, but to also help models disambiguate between the different "senses" of a word, or in other words, the different meanings of the same word in different context. For example, "along" in Spatial-STUPD has two meanings in different contexts. Similarly in Temporal-STUPD, although "before" is a subset of "by", they have different subtle meanings. For example, "before" can be used to depict a relation between two events (event A before event B), or an event and a time point (Event A before Time B), but "by" is explicitly used to depict a relation between an event and a time point (Event A by Time B). (You also mentioned "Also, I don't think "by" (which implies a deadline) is used to describe the timing of two events in general". This is true, and has already been factored in the paper. See Figure 2 in the paper).
>
> 3. " Clarity. In the #1 callout in Sec 3.4.1, the author pointed out that 'track' and 'tunnel' have fewer relationships than other object types.". This design is quite very intentional, and does not arise out of limitations in UNITY. As we mentioned in the paper, we follow a structured object-pair selection process to generate the dataset. This involves filtering out object-pairs that are unlikely to occur in the real world, and also a size constraint (both objects in the pair should have similar scales of size, otherwise the resulting images would not be able to capture both objects without affecting the visibility of at least on the objects in the object pair). Due to this, tunnels and track (being large scale objects) are paired with only other larger objects (like vehicles, furniture, buildings, etc). The other reason for the lesser frequency of tunnel/track is that in the spatial relation triplet (Object A, relation, Object B), tunnels and tracks are semantically restricted to Object B. It is unlikely to see tunnel and track being the subject of a sentence (eg. a track is above a car is not only logically inconsistent, but also semantically incorrect). This is why, overall, we see lesser occurrences of these two types of objects.
>
>
> [Minor corrections]
> 4. citation format. Thanks for pointing this out. We have rectified this now.
> 5. "The author frequently used "<" and ">" without any escape character." This is corrected too.
> 6. "2nd line of 2nd para of Intro. I cannot understand what pre is for. ". This was a wrong citation, and has now been corrected.
> 7. formating of large numbers with Commas. Corrected.
>
> [1] Xiao, J., Shang, X., Yao, A., & Chua, T. S. (2021). Next-qa: Next phase of question-answering to explaining temporal actions. In Proceedings of the IEEE/CVF conference on computer vision and pattern recognition (pp. 9777-9786).
>
> [2] Hendricks, L. A., Wang, O., Shechtman, E., Sivic, J., Darrell, T., & Russell, B. (2018). Localizing moments in video with temporal language. arXiv preprint arXiv:1809.01337.

---

### Official Review · Reviewer_ztaS · 2023-11-01

**Soundness:** 3 good
**Presentation:** 3 good
**Contribution:** 3 good
**Rating:** 5
**Confidence:** 5

**Summary:**

This papers proposes STUPD dataset, which addresses the lack of diversity in prepositions in existing datasets and the absence of dynamic prepositions that involve motion. It consists of 150K images and videos capturing 30 spatial relations and 50K video sets depicting 10 temporal relations, all with 3D info and bounding box annotations. Authors claim that pre-training on STUPD can significantly improve performance on real-world visual reasoning tasks.

**Strengths:**

1. Addresses a gap in existing datasets by including a wider variety of prepositions and introducing dynamic prepositions.
2. Provides a synthetic dataset with both spatial and temporal relations, which is crucial for a more holistic understanding of visual reasoning.
3. Demonstrates the real-world applicability of the dataset through pre-training improvements on visual reasoning tasks.
Weaknesses:

**Weaknesses:**

1. The synthetic data may not fully capture the complexity of real-world scenarios.
2. The paper could benefit from a more extensive validation of the dataset's efficacy across a broader range of models and tasks.

**Questions:**

1. How well do models trained on STUPD perform when applied to real-world data, considering the dataset is synthetic?
2. How does STUPD handle ambiguous or context-dependent prepositions where the spatial or temporal relationship might not be clear-cut?
3. What measures are in place to ensure that the synthetic data in STUPD is diverse and representative of real-world scenarios?

---

> ### Author Response · Authors · 2023-11-21
> **Viability of a synthetic dataset**
>
> Dear Reviewer, thanks for your comment. Understandably, your concerns are valid. We had this concern right from the beginning, and took many steps to ensure that a synthetic dataset is useful to real world applications as well.
>
> 1. "The synthetic data may not fully capture the complexity of real-world scenarios."
> We ensured that the selection of objects, and the nature of interaction was as realistic as possible. Not only did we use textured real-world-like objects, but restricted object pairs to only those that are likely to be seen in the real world. Also the physics engine used ensures that the interaction is natural.
>
> 2. "The paper could benefit from a more extensive validation of the dataset's efficacy across a broader range of models and tasks."
> Agreed. We realize that there could be more experiments for the temporal part of STUPD. So we will include some results along those lines as well.
>
> 3. "How well do models trained on STUPD perform when applied to real-world data, considering the dataset is synthetic?"
> We demonstrated through the pretraining experiments (Table 3 and 4), that when models are first pretrained on STUPD, and then fine tuned on real-world datasets, the model performance is. consistently better.
>
> 4. "How does STUPD handle ambiguous or context-dependent prepositions where the spatial or temporal relationship might not be clear-cut?"
> Different context dependent prepositions have distinct labels and videos corresponding to them. For example, the word "against" has two meanings, and different corresponding videos. Similarly, there are two different classes corresponding to "into". Within certain constraints, we cover all prepositional relations in our dataset. Maybe some less-frequently words are not covered, but their meaning is covered by one or the other of the classes in this dataset. There are 40 classes in total, and the full list can be seen in Appendix.
>
> 5. "What measures are in place to ensure that the synthetic data in STUPD is diverse and representative of real-world scenarios?"
> We started out by selecting a diverse range of objects from various supercategories (like vehicles, large grounded objects, small objects, larger objects, containers). Overall, the diversity of objects as well as the absolute number of objects is comparable to SpatialSense (a real world dataset). These synthetic objects have real-world-like textures. We also filter out object pairs that are unlikely to be seen in the real world (such as building on top of a car).
> Backgrounds are also selected so as to bring realness. Finally, a physics engine ensures that the interactions are real-world-like.

---

### Meta-Review · Area_Chair_zdHj · 2023-12-09

**Metareview:**

The paper introduces the Spatial and Temporal Understanding of Prepositions Dataset (STUPD), a large-scale video dataset for understanding static and dynamic spatial and temporal relationships, enhancing computer vision models' ability to detect visual relationships, evidenced by improved performance in real-world datasets. All the reviewers recommend rejection of the paper, while they all agree this is an interesting direction, the common worry is that the dataset is synthetic and there is a large gap to the real-world scenes. After carefully reading the paper and the rebuttal, the AC agrees with the reviewers.

**Justification For Why Not Higher Score:**

All the reviewers recommend rejection of the paper, while they all agree this is an interesting direction, the common worry is that the dataset is synthetic and there is a large gap to the real-world scenes.

**Justification For Why Not Lower Score:**

N/A

---

### Decision · Program_Chairs · 2024-01-16

Reject